# Electropolymerized, Molecularly Imprinted Polymer on a Screen-Printed Electrode—A Simple, Fast, and Disposable Voltammetric Sensor for Trazodone

**DOI:** 10.3390/s22072819

**Published:** 2022-04-06

**Authors:** Isabel Seguro, Patrícia Rebelo, João G. Pacheco, Cristina Delerue-Matos

**Affiliations:** REQUIMTE/LAQV, Instituto Superior de Engenharia do Porto, Instituto Politécnico do Porto, Rua Dr. António Bernardino de Almeida 431, 4200-072 Porto, Portugal; mgsps@isep.ipp.pt (I.S.); patricia.rebelo@graq.isep.ipp.pt (P.R.); cmm@isep.ipp.pt (C.D.-M.)

**Keywords:** trazodone, antidepressant drugs, electrochemical sensor, molecularly imprinted polymer, disposable analytical devices

## Abstract

In recent years, analytical chemistry has been facing new challenges, particularly in developing low-cost, green, and easy-to-reproduce methods. In this work, a simple, reproducible, and low-cost electrochemical (voltammetric) molecularly imprinted polymer (MIP) sensor was designed specifically for the detection of trazodone (TZD). Trazodone (TZD) is an antidepressant drug consumed worldwide since the 1970s. By combining electropolymerization (surface imprinting) with screen-printed electrodes (SPCEs), the sensor is easy to prepare, is environmentally friendly (uses small amounts of reagents), and can be used for in situ analysis through integration with small, portable devices. The MIP was obtained using cyclic voltammetry (CV), using 4-aminobenzoic acid (4-ABA) as the functional monomer in the presence of TZF molecules in 0.1 M HCl. Non-imprinted control was also constructed in the absence of TZD. Both polymers were characterized using CV, and TZD detection was performed with DPV using the oxidation of TZD. The polymerization conditions were studied and optimized. Comparing the TZD signal for MIP/SPCE and NIP/SPCE, an imprinting factor of 71 was estimated, indicating successful imprinting of the TZD molecules within the polymeric matrix. The analytical response was linear in the range of 5–80 µM, and an LOD of 1.6 µM was estimated. Selectivity was evaluated by testing the sensor for molecules with a similar structure to TZD, and the ability of MIP/SPCE to selectively bind to TZD was proven. The sensor was applied to spiked tap water samples and human serum with good recoveries and allowed for a fast analysis (around 30 min).

## 1. Introduction

Depression is a mental disorder, classified as a mood disorder, with a significant impact on the quality of life of both the persons who have this illness and those closest to them [1]. It is a major cause of disability, [2] affecting more women and the elderly, [1] and has a diverse distribution all over the world [3]. An estimated 5% of the adult population suffers from depression worldwide [4]. It is associated with diseases with symptoms linked to pain and inflammation [5], closely related to suicide [6] and inadequate general health [3]. Different antidepressant drugs are accessible for the treatment of depression, with various action mechanisms.

Trazodone (TZD) is an antidepressant drug that belongs to the class of serotonin antagonist and reuptake inhibitor [2,7,8,9]. It is a triazolopyridine derivate and presents antidepressant, anxiolytic, and hypnotic effects, but produces lower anticholinergic side effects than other antidepressants [10]. TZD was first authorized in the 1970s for the treatment of depression. However, due to its heterogeneous mechanism, other therapeutic uses have been reported, namely, sleep disturbance [11], anxiety disorders, dementia, psychosis, parkinsonism [12], trembling states, emotional disorders, or dyskinesia [13]. Some side effects are detailed for this antidepressant drug, including nausea, insomnia, agitation, dry mouth, constipation, headache, hypotension, blurred vision, and confusion [11]. Hepatotoxicity is another adverse effect reported for TZD, mainly by oxidative stress and by dysfunction in intracellular organelles [13,14]. Additionally, interactions with other therapeutic drugs are reported, namely for TZD and warfarin [15], and TZD and tandospirone [16].

The monitorization of therapeutic drugs is useful both in environment quality control, since they are not completely metabolized after consumption, and wastewater treatment plants were not designed for their removal [7,8,17], and for clinical purposes, for therapeutic compliance or the management of therapeutic levels.

Several analytical methods have been described for TZD determination in different matrices, namely, potentiometric [9], voltametric [17], atomic absorption and emission methods [18], polarography [19], HPLC [20], GC [21], spectrophotometric and spectrofluorimetric [22], electrochemical detection including MWCNT-modified glassy carbon electrodes [11], and trazodone-selective electrodes [13]. Additionally, for the electrochemical determination of TZD, some potentiometric sensors with different ion-exchange materials and ion pairs as electroactive materials have been reported [23].

In recent years, efforts have been made in analytical chemistry in different areas, such as environmental pollution and health monitoring, trying to develop alternatives to traditional analytical methods which often involve complex and expensive instrumentation, the high consumption of organic solvents, long analysis times, specialized analysts and, sometimes, expensive and laborious sample preparations.

Due to their characteristics, namely, simplicity, sensitivity, low cost, and the possible integration into portable devices and automatic systems, electrochemical sensors have proved to be good analytical tools for fast and in situ analysis. The use of molecularly imprinted polymer (MIP)-based electrochemical sensors has attracted attention. In comparison with commonly used natural recognition elements (antibodies, enzymes, and aptamers), MIPs are more robust and resistant to chemical and thermal conditions, stable, easy, and cheap to prepare. Molecularly imprinted technology provides selective sensors since they are built with specific recognition sites for the molecule that we want to analyze. MIPs for pharmaceuticals have been designed and have proved to be an excellent choice for the construction of highly selective sensors [24,25,26,27,28,29].

MIPs are synthetic materials, built to create recognition cavities for a specific molecule. As such, a template, the analyte, is surrounded by a selected functional monomer in a pre-polymerization solution. The synthetic polymer shall be complementary to the analyte in shape, size, and functional groups, so, theoretically, it is possible to build a suitable MIP for any molecule. There are several ways to achieve polymerization, namely by bulk polymerization, precipitation, or electropolymerization [30]. In pharmaceutical analysis, both polymerizations have been reported [24,25]. For electrochemical sensor production, electropolymerization has proved to be a convenient approach, since the polymer will grow on the working electrode surface without the need for an additional step of immobilization. Furthermore, it allows for easy control of the polymer thickness by controlling the current, and it is a fast and easy-to-reproduce technique. After the removal of the template from the MIP, the sensor is ready to be used. By combining it with screen-printed carbon electrodes (SPCEs), a portable, small-sized, and disposable sensor can be constructed which can be integrated into commercial portable devices. Several models of portable potentiostats are already available on the market from DropSens and PalmSens, smartphone compatibility. Although there are numerous reports of MIP sensors for pharmaceutical detection [24,25,26,31,32,33], to the best of our knowledge, there are no reports of a TZD MIP sensor.

So, in the present work, MIP technology was used for the first time to develop a sensor for the electrochemical detection of trazodone. The selective voltammetric MIP sensor was constructed via direct surface imprinting of the antidepressant drug onto the surface of a commercial screen-printed carbon electrode (SPCE), through electropolymerization of a solution containing 4-ABA (in accordance with the findings in preliminary tests of several electropolymers) and trazodone. The sensor was tested in spiked water and human serum samples. This is the first report of an MIP sensor for TZD. Additionally, SPCEs are suitable for point-of-care analysis and environmental monitorization due to their reduced size. The chemical structure of TZD and 4-ABA are presented in Figure 1.

## 2. Materials and Methods

### 2.1. Reagents

Trazodone hydrochloride and 4-aminobenzoic acid were purchased from Sigma-Aldrich (St. Louis, MO, USA), and both standard solutions were prepared by dissolving the appropriate amount in 0.1 M hydrochloric acid (HCl) from Fluka. Sulfuric acid (Merck, Kenilworth, NJ, USA), methanol (Romil, Cambridge, UK), acetic acid (Carlo Erba, Emmendingen, Germany), lansoprazole (Sigma), atenolol (Sigma), and fenofibrate (sigma) were also used. All commercial reagents were of analytical grade and used without purification. Through this work, ultra-pure water (resistivity = 18.2 MΩcm), obtained from a Millipore Simplicity 185 water purification system, was used.

### 2.2. Apparatus

Voltammetric measurements were taken using an Autolab PGSTAT 101 potentiostat–galvanostat with NOVA 1.10 software (Metrohom). For electrochemical impedance, spectroscopic analysis was employed, using an Autolab PGSTAT128N controlled by NOVA 1.6 (Metrohom, Herisau, Switzerland). Commercial screen-printed carbon electrodes, from Dropsens DRP-110, with carbon working (d = 4.0 mm) and auxiliary electrodes and a silver pseudo-reference electrode, were used.

### 2.3. MIP Sensor Fabrication

Initially, a bare SPCE was activated with 0.5 M H_2_SO_4_ and applying cyclic voltammetry (CV) for 10 scans between −0.2 and 1.2 V with a 100 mV s^−1^ scan rate. Electropolymerization was then performed using CV (20 cycles, scan rate 100 mV s^−1^, between −0.2 and 1.4 V) in the presence of 50µL of a polymerization solution, with 5 mM of 4-ABA and 2.5 mM of TZD in 0.1 M HCl. TZD-entrapped molecules were removed placing a 0.5 M H_2_SO_4_ solution on the working electrode for 60 min. A control non-imprinted polymer (NIP) was also prepared with the same procedure, yet without TZD.

### 2.4. Electrochemical Measurements

The determination of TZD was performed using differential pulse voltammetry (DPV), between 0.0 V and 1.6 V at a scan rate of 0.05 V/s, using 0.1 M HCl as the supporting electrolyte. After a 30 min incubation period, 10 µL TZD solutions were placed on the sensor. CV was used for MIP/NIP characterization throughout the construction process, placing 40 µL of 0.5 mM [Fe(CN)_6_]^3−/4−^ in 0.1 M KCl on the SPCE. CV was performed with a 100 mV/s scan rate. Measurements were carried out at room temperature.

### 2.5. Samples Analysis

Human serum and tap water were tested. The samples were initially acidified with concentered HCl. This step was executed since the previous experiments were performed with TZD prepared in 0.1 M HCl solution. No further pre-treatment was used. Then, samples were spiked with TZD, and the analysis was performed under the conditions described above.

## 3. Results and Discussion

### 3.1. Molecularly Imprinting of TZD by Electropolymerization

Electropolymerization was chosen for MIP sensor preparation due to its characteristics, namely, simplicity, control of the polymerization layer, preparation of a selective recognition element (MIP), and attachment to the transducer (carbon electrode) in one single step. Typically, this procedure is performed in aqueous solutions to avoid the use and waste of organic solvents. The steps involved in the construction of the sensor are illustrated in Figure 2.

Preliminary tests for functional monomer selection were performed, where several electroactive monomers (pyrrole, aniline, phenol, and 4-ABA) were tested. 4-ABA was selected since it proved able to bind to TZD molecules, while no binding was observed with the other monomers. This is because, generally, non-covalent interaction between monomer (4-ABA) and template (TZD) was used. Thus, 50µL of a pre-polymerization solution containing 5 mM 4-ABA and 2.5 mM TZD in 0.1 M HCl was placed on the surface of a commercial screen-printed carbon electrode. Electropolymerization was then performed using cyclic voltammetry for 20 scans, between −0.2 V and +1.4 V. A polymeric film with entrapped TZD molecules was produced (MIP). The same procedure was performed for a control non-imprinted polymer (NIP) but without the presence of TZD. Figure 3A,B show the layer-by-layer formation of the polymers through electropolymerization, with the oxidation peak of 4-ABA decreasing with the number of cycles. However, clear differences between the NIP (Figure 3A) and MIP (Figure 3B) polymerizations were registered. While no significant differences were found during the first polymerization cycle, with the increase in cycles, the peaks of TZD can be seen in MIP polymerization. Otherwise, the behavior of the 4-ABA peaks is also different between MIP and NIP.

To confirm the success of TZD incorporation within the polymer, DPV measurements were taken after polymerization. The electrodes were washed with water, dried, and then analyzed via DPV with HCl 0.1 M as the electrolyte solution. The results are expressed in Figure 3C. As it can be seen, the NIP/SPCE voltammogram does not show any peaks, while two oxidation peaks for TZD can be observed in the MIP/SPCE voltammogram. These results indicate the success of the imprinting of TZD in MIP preparation.

### 3.2. Electrochemical Behavior of TZD at MIP/SPCE

After the successful incorporation of the template (TZD), the molecules should be extracted to create specific cavities, and then, the ability of the sensor to selectively rebind should be evaluated. The extraction is one of the most difficult steps of MIP preparation. A complete template removal should be assured. This can be confirmed using DPV measurements until the TZD peak completely disappears. The TZD molecules were extracted by placing a 0.5 M H_2_OS_4_ solution on the modified working electrode. It was concluded that no peak of TZD was registered on DPV voltammograms after 60 min of extractions (Figure 4). These results show the success of TZD extraction under those conditions. Then, the rebinding of TZD molecules in both the NIP/SPCE and the MIP/SPCE was tested. A 10 µM TZD standard solution was prepared in HCl 0.1 M and placed on the surface of the sensors. After 30 min of incubation, the sensors were washed with water, dried, and DPV was finally executed in HCl 0.1 M to confirm the binding of the TZD molecules. Figure 3 shows the obtained voltammograms. An oxidation peak of TZD was observed for both NIP/SPCE and MIP/SPCE, confirming the ability of the poly-4-ABA to bind to TZD molecules. It can also be observed that the MIP/SPCE voltammetric signal is noticeably higher than the NIP/SPCE signal. The binding in NIPs is common and is usually attributed to non-specific binding. The larger the difference between the NIP and MIP signal, the better the imprinting. This can be evaluated by estimating the imprinting factor (ip current MIP/ip current NIP). Using these results, an imprinting factor 71 was obtained. The higher voltametric peak at MIP/SPCE is due to the successfully created specific binding sites, allowing for selective binding to TZD molecules.

### 3.3. Optimization of Experimental Conditions

When using electropolymerization, there are several experimental parameters that should be tested and optimized since they influence the MIP’s efficiency. Initially, polymerization conditions, namely, the concentration of monomer (4-ABA), the concentration of template (TZD), and the number of polymerization cycles, were optimized. Then, the extraction and incubation conditions were also studied.

#### 3.3.1. Polymerization Conditions

Initially, the optimization of the functional monomer concentration during polymerization was performed. Different sensors were prepared using distinct concentrations (1, 5, 10, 15, and 20 mM) of the monomer 4-ABA in the polymerization step, while maintaining the concentration of 2.5 mM for TZD, 15 cycles for polymerization, and 10 min for incubation. After extraction and subsequent incubation with TZD (100 µM), a better response of the MIP/SPCE sensor for 5 mM of 4-ABA was observed (Figure 5). An increase from 1 to 5 mM of 4-ABA was observed, but then, the peak current decreased for 15 mM, and no oxidation peak was observed for 20 mM. An explanation for this behavior may be related to the increase in polymer thickness with the 4-ABA concentration, making the TZD molecules too entrapped to be efficiently extracted. On the other hand, this could also be explained by a much higher proportion of polymer than in the template, which may not be good for cavity formation, leading to ineffective binding. NIP/SPCE sensors were also tested, with no variation in the analytical response.

Subsequently, the template (TZD) concentration during polymerization was also studied. Keeping all other analysis conditions unchanged and 5 mM 4-ABA in the polymerization, the TZD concentration was tested from 0.1 to 5 mM. An increase in the peak current (Figure 5B) up to 2.5 mM was found, followed by a decrease when using 5 mM. With the increase in TZD concentration, more molecules were imprinted, and more specific binding sites were created. However, for higher concentrations, the TZD molecules could be too entrapped in the polymer and the cavities could not be effectually produced. Therefore, 2.5 mM of TZD was chosen for MIP/SPCE preparation.

One of the major advantages of the electropolymerization technique is the ability to control the thickness of the polymeric film formed on the surface of the electrode. This can be achieved by changing the number of CV scans during polymerization. Therefore, scans between 5 and 30 were tested for both NIP/SPCE and MIP/SPCE, followed by TZD analysis, keeping 2.5 mM TZD and 5 mM 4-ABA for the polymerization solution, 60 min for extraction, and 10 min for incubation. The results are expressed in Figure 5C. For NIP/SPCE, the analytical response showed no significant variation with different cycles and very little response after incubation. However, for MIP/SPCE, a variation in the analytical signal with the number of cycles can be observed. There is a significant increase in the peak current intensity from 5 to 20 cycles. Then, there is a stabilization in the signal, probably because at this point the polymer thickness might be too great for the entrapped TZD molecules to be easily extracted and rebound. Thus, 20 cycles were selected for the following studies.

#### 3.3.2. Incubation Time

Another important factor evaluated was the incubation time, since adsorption of TZD molecules to the sensor is directly related to this parameter. For the parameters, with 2.5 mM TZD, 5 mM 4-ABA, 20 cycles, and 60 min for extraction maintained, 40 µL of a 10 µM TZD solution was placed on the surface of the sensor for 5, 10, 30, and 60 min; then, the solution was removed and DPV was performed with HCl 0.1 M as the supporting electrolyte. The obtained results are expressed in Figure 6. The intensity of the peak current increases with time until 30 min. However, there is a decrease for longer incubation times, so 30 min was chosen for incubation.

#### 3.3.3. Extraction Conditions

The extraction step is one of the critical points of MIP preparation, since complete template removal must be achieved. So, the study of the optimal extraction solutions and conditions is fundamental. In this work, the effectiveness of different solvents in the extraction of TZD molecules from the poly 4-ABA matrix was tested. As such, 0.1 M phosphate buffer at pH = 7, MeOH, 0.5 M H_2_SO_4_, 0.1 M NaOH, and H_2_O were selected. The extraction was considered complete when no oxidation peak of TZD was observed in DPV analysis. After extraction with the tested solvents, the incubations and DPV analysis of TZD were performed. As can be observed from Figure 7, significant differences were revealed. Although all the other solvents were able to extract TZD molecules, the best incubation results were achieved after extraction with 0.5 M H_2_SO_4_.

### 3.4. Characterization of the Construction of the Sensor

A step-by-step characterization of SPCE modifications was performed with CV using a solution of 0.5 mM [Fe(CN)_6_]^3−/4−^ in 0.1 M KCl solution. This can be used to study the changes in the working electrode surface and to confirm the formation of the polymeric films. As it can be seen in Figure 8, using unmodified SPCEs, two well-defined redox peaks were registered. The analysis of MIP and NIP shows significant differences, not only from SPCE but also between NIP and MIP, showing that polymers with different characteristics were formed. For NIP, no redox peaks were obtained. For MIP, it is possible to observe some redox activity, which may indicate that the presence of TZD molecules leads to the formation of the specific cavities. Finally, after TZD incubation, there is a small decrease in the redox peaks, demonstrating the rebinding of TZD molecules to the polymeric film.

### 3.5. Analytical Performance

The determination of different TZD concentrations in the MIP/SPCE sensor with DPV was performed using the optimized experimental conditions. A linear relationship was observed between TZD concentration and the peak current in the range between 5 and 80 µM (Figure 9B). Experiments were performed in triplicate. The obtained calibration curve had the following analytical parameters: r^2^ = 0.9972, ip(μA) = 0.2346 × [TZD](μM) −0.7927. The limit of detection (LOD) and quantification (LOQ) were estimated to be 1.6 µM and 5.4 µM, respectively, given by equations: LOD = 3 s/m and LOQ = 10 s/m, where “s” is the standard deviation of the intercept and “m” is the slope of the calibration plot. Each sensor was re-used three times without a significant change in the current response. In Table 1, a comparison with other electrochemical methods is presented.

### 3.6. Selectivity Studies

Selectivity is a key feature of an MIP sensor. The ability of the constructed MIP/sensor to selectively recognize TZD was assessed by comparing the response to TZD with the response to two drugs with similar structures, atenolol (ATN) and lansoprazole (LNS). A solution of 0.1 mM of TZD was analyzed, while for the analogue structures, solutions with 10 mM (100×) were analyzed. The electrochemical response for both NIP/SPCE and MIP/SPCE was evaluated. The solutions of each compound were analyzed individually, and the results were compared (Figure 10). The results were expressed as the peak current obtained. It can be concluded that the MIP/sensor is highly selective to TZD molecules and can easily discriminate the analogue structures evaluated even at 100× higher concentrations. For LNS, no oxidation peaks were found for both NIP/SPCE and MIP/SPCE. For ATN, a very low oxidation peak was observed at MIP/SPCE. The combination of a very high imprinting factor with the huge difference of signals to the analogue molecules proves the successful construction of an MIP sensor with highly specific binding sites for TZD molecules.

### 3.7. Application to Real Samples

In order to validate the application of the proposed MIP sensor in different matrices, after optimization, the sensor was used for the analysis of TZD in tap water, collected in the lab, and human serum (from human male AB plasma sterile-filtered). Both samples were spiked with two different concentrations of TZD (5 and 10µM for water; 10 and 100 µM for human serum) after adjusting pH and diluting with HCl 0.1 M. The standard addition method was used for the quantifications. Table 2 summarizes the obtained results. It was possible to quantify TZD in the samples with the constructed MIP sensor. Recoveries between 94 and 101% were found for the tap water sample, and between 71 and 78% for human serum, probably due to matrix effects. These results show that the proposed sensor can be used in different samples with simple pre-treatment steps.

## 4. Conclusions

For the first time, an MIP electrochemical sensor for the detection of the antidepressant drug TZD has been reported. By combining eletropolymerization with SPCE electrode technology, the preparation procedure is very simple, easy to reproduce, has a low cost, and is environmentally friendly. The MIP sensor is disposable and can be integrated into portable analytical devices for in situ analysis. It shows good analytical performance with good sensitivity, a highly elevated imprinting factor, and the ability to specifically distinguish molecules with analogue structures. Furthermore, it was successfully applied in the determination of TZD in two different real samples, water and human serum. The proposed MIP sensor shows great potential to be mass produced and to be integrated with portable devices for TZD detection.

## Figures and Tables

**Figure 1 sensors-22-02819-f001:**
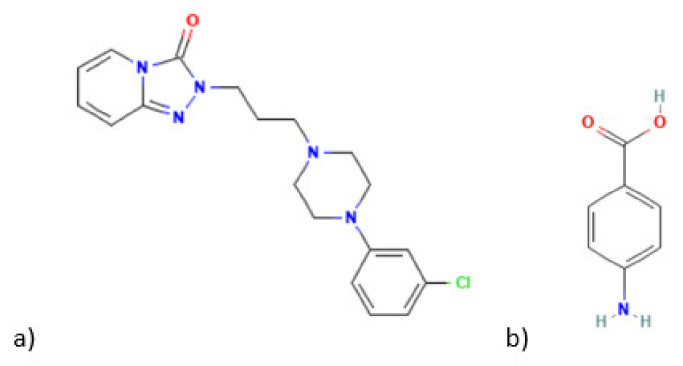
(**a**) Trazodone, (**b**) 4-aminobenzoic acid.

**Figure 2 sensors-22-02819-f002:**
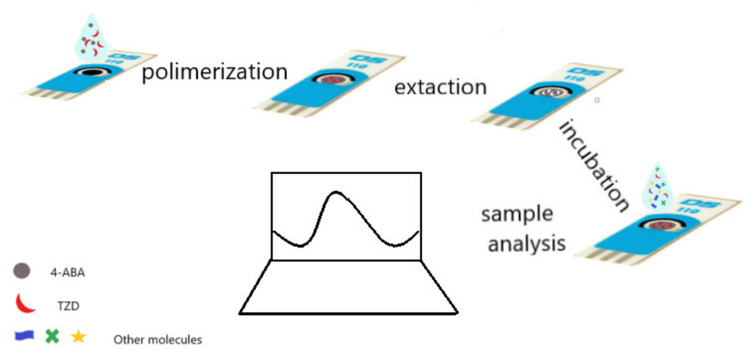
Schematic illustration of the preparation of the MIP/SPCE sensor.

**Figure 3 sensors-22-02819-f003:**
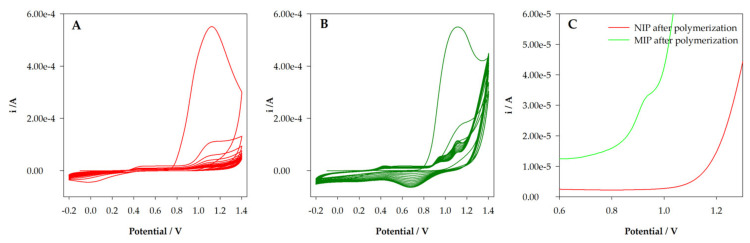
(**A**) Electropolymerization of NIP from a solution containing 5 mM 4-ABA in 0.1 M HCl; (**B**) electropolymerization of MIP from a solution containing 5 mM 4-ABA and 2.5 mM TZD in 0.1 M HCl; (**C**) DPV analysis of NIP and MIP sensors after polymerization.

**Figure 4 sensors-22-02819-f004:**
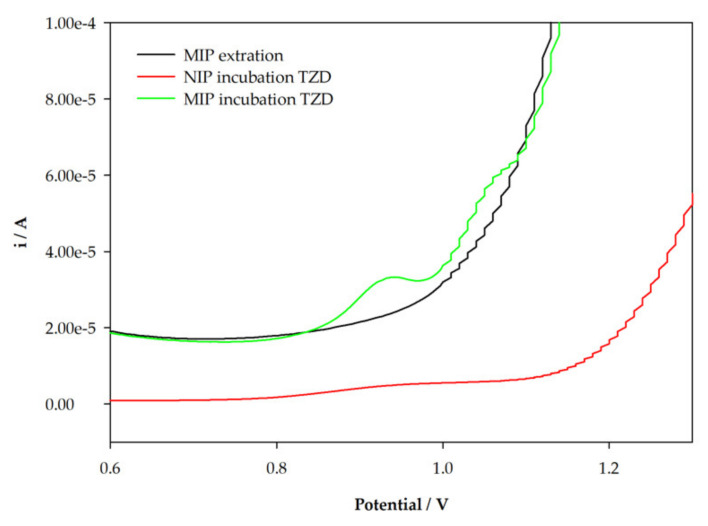
DPV voltammograms after incubation of TZD solution.

**Figure 5 sensors-22-02819-f005:**
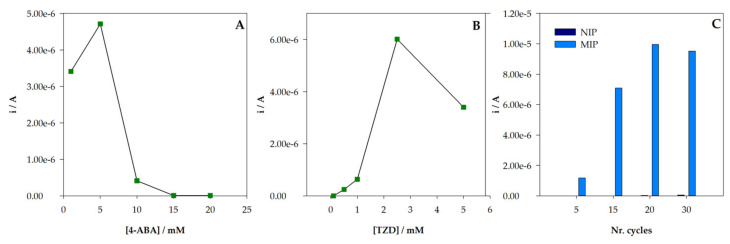
Optimization of the MIP preparation conditions: (**A**) variation of the peak current intensity with the concentration of monomer 4-ABA; (**B**) variation of the peak current intensity with the concentration of template TZD; (**C**) variation of the peak current intensity with number of CV cycles.

**Figure 6 sensors-22-02819-f006:**
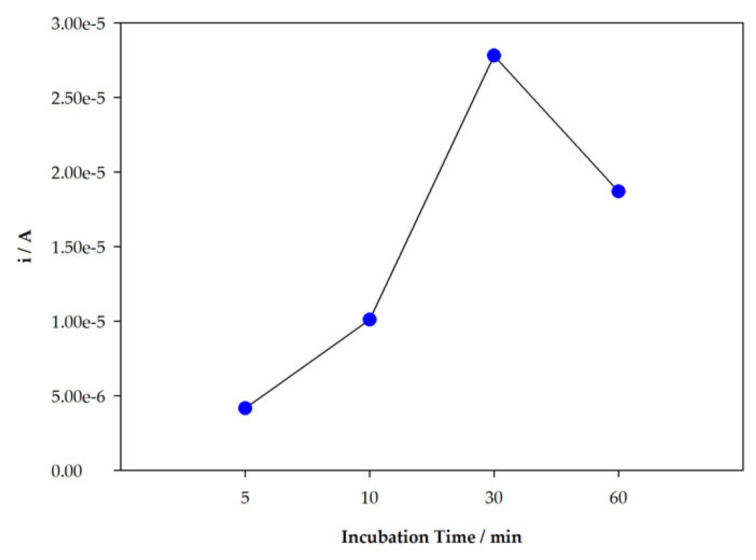
Variation of the peak current intensity with incubation time.

**Figure 7 sensors-22-02819-f007:**
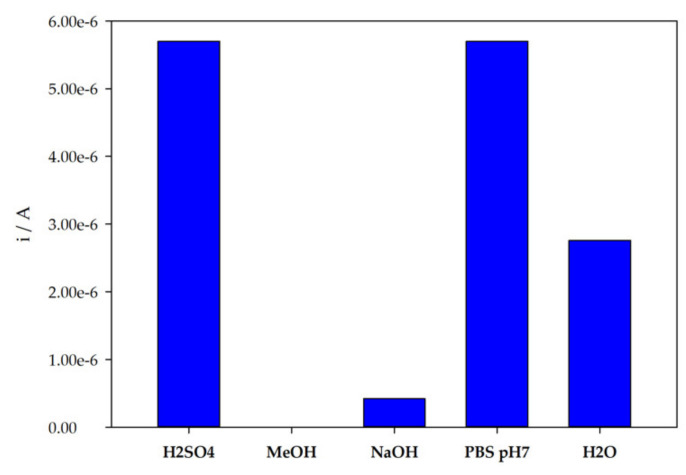
Variation of the peak current intensity after extraction and with different solutions and rebinding to TZD.

**Figure 8 sensors-22-02819-f008:**
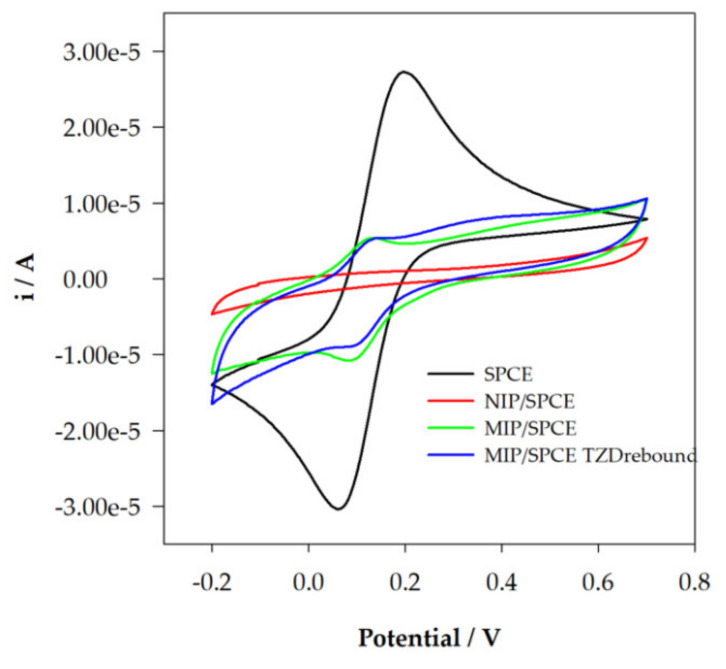
Sensor’s construction characterization with CV 0.5 mM [Fe(CN)_6_]^3−/4−^ in 0.1 M KCl.

**Figure 9 sensors-22-02819-f009:**
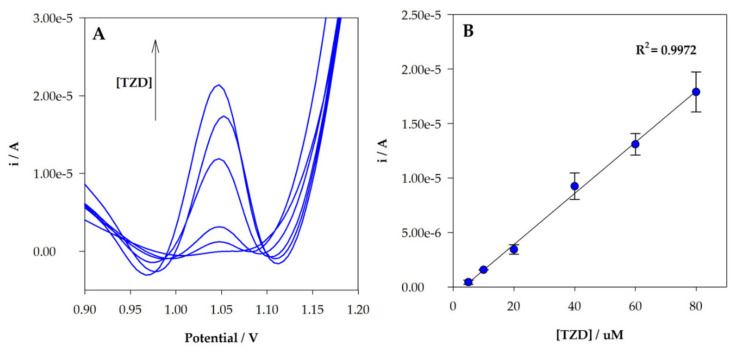
(**A**) DPV voltammograms obtained for MIP sensor analysis of different TZD concentrations; (**B**) linear relationship between peak current intensity and TZD in the concentration range 5 to 80.0 µM.

**Figure 10 sensors-22-02819-f010:**
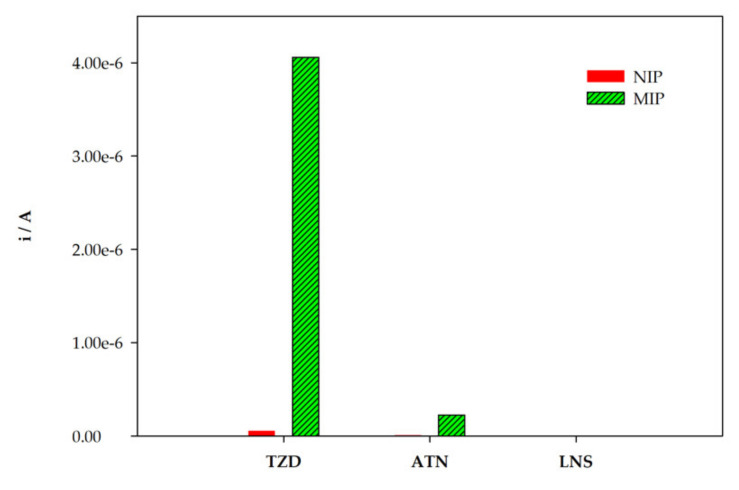
Selectivity studies.

**Table 1 sensors-22-02819-t001:** Comparation with other electrochemical methods.

Analytical Method	Samples	LOD	References
Potentiometry	Urine Pharmaceutical preparations	1.8 × 10^−5^ M	[13]
Voltammetry	Pharmaceutical preparations	1.7 × 10^−6^ M	[34]
Voltammetry	Human serum water	1.6 × 10^−6^ M	This work

**Table 2 sensors-22-02819-t002:** Determination of DCF concentration in spiked water samples with the constructed MIP sensor.

Sample	[TZD]_add_, µM	[TZD]_det_, µM	Recovery, %
Tap water	0		
	5	4.68	94
	10	10.1	101
Human serum	0		
	10	7.06	71
	100	77.8	78

[TZD]_add_: TZD concentration added to the sample; [TZD]_det_: TZD concentration determined in the sample by standard addition method, with the sensor presented in this work.

## Data Availability

Not applicable.

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
