# Peer review of "Electropolymerized, Molecularly Imprinted Polymer on a Screen-Printed Electrode—A Simple, Fast, and Disposable Voltammetric Sensor for Trazodone"

_sensors, 2022, doi:10.3390/s22072819_

Round 1
Reviewer 1 Report
This manuscript describes the combination of MIP sensor with SPCE electrodes technology to detect antidepressant drug TZD. Authors chose electro-polymerization for control of the layer cycle, preparation of MIP, and attachment to the transducer in one single step. It seems that the optimization experiment methods are adequately described and detection in real samples (tap water and human serum) results were well performed.
I do recommend this manuscript with minor revision:
- The authors should provide Figures with good resolution. It would be better if there is a more specific scheme or picture for the sensor.
- There are a few typos including in the caption of Fig 2.
Reviewer 2 Report
- The acronyms used through the manuscript, such as CV, DPV, and EIS somewhere should be defined. Assuming that EIS stands for Electrochemical Impedance Spectroscopy, no EIS data are discussed and reported in the manuscript.
- Reference [7] and [16] are not cited in the manuscript.
- The authors should explain why “It seems that eletropolymerization of MIP indicates the formation of a high surface area, probably due to the incorporation of TZD molecules” (rows 266-268).
- The authors should mark, in the MIP/SPCE voltammogram of Fig. 2c, the positions of the two oxidation peaks for TZD they refer to in the manuscript (rows 172-173).
- Caption of fig. 2b, row 122 should be corrected.
- Blank spaces should be eliminated (rows 190, 210, 230, 280, 306).
- I understand that the properties of modified screen-printed electrodes, prepared using different polymerization conditions, with different incubation time, and extraction conditions, have been compared to optimize the performances of the electrodes. The results shown in Figs. 4 refer to the way “the peak current intensity” changes with the concentrations of the monomer, the concentration of TZD, and the increasing number of cycles. Fig. 5 shows how the “peak current intensity” changes with the incubation time. Fig. 6 (the caption of which should be corrected) reports the “peak current intensity” as a function of the solvent used to extract TZD from the electrodes. For a better understanding, as only one out of five parameters affecting the electrode’s behavior is changed and reported on the horizontal axis, an inset could be added to each figure, reporting the values of parameters that are set constant. In addition, it is not clear to me what current peak do the authors refer to: is it one of the couple of peaks observed in the green line of Fig. 3? Which one? Information concerning the monomer and TZD concentration of the sample to which Fig 3 refers to, should be added.
- In section 3.5, while discussing the linear response of the sensor to the analyte reported in Fig. 8b, the authors refer to figure 7 (rows 289-290).
- In section 3.6, while discussing the selectivity of the sensor to TZD compared to other analytes, evidenced by Fig. 9, the authors refer to figure 8 (rows 306-307).
Reviewer 3 Report
The authors described the detection of trazodone using an electrochemical sensor based on MIP.
The manuscript presents a decent English level, but there are many small mistakes and some unclear phrases in the text (see attached document). Please correct them.
The study was properly described, with some possible improvements. In the attached document you have my remarks and suggestions for improving the manuscript. For these reasons, I recommend the publication of this manuscript only after minor revision.
